# Factors Associated with Acceptance of Vaccination Against Human Papillomavirus in eThekwini District of South Africa

**DOI:** 10.3390/vaccines13070732

**Published:** 2025-07-06

**Authors:** Phelele Bhengu, Charles S. Wiysonge, Patrick D. M. C. Katoto, Duduzile Ndwandwe, Sara Cooper, Sebenzile Bhengu, Akhona V. Mazingisa, Theresa Saber, Mandisi Sithole, Darian Smith, Lindiwe G. Tembe, Paul Kuodi, Muki S. Shey

**Affiliations:** 1Department of Medicine & CIDRI-Africa, Faculty of Health Sciences, University of Cape Town, Cape Town 7700, South Africa; bhnphe001@myuct.ac.za (P.B.); muki.shey@uct.ac.za (M.S.S.); 2Cochrane South Africa, South African Medical Research Council, Cape Town 7505, South Africa; katoto.chimusa@ucbukavu.ac.cd (P.D.M.C.K.); duduzile.ndwandwe@mrc.ac.za (D.N.); sara.cooper@mrc.ac.za (S.C.); ezamashongololo@gmail.com (S.B.); 21213054@dut4life.ac.za (A.V.M.); proudlysa@msn.com (T.S.); mandisisithole2@gmail.com (M.S.); gracelindiwe95@gmail.com (L.G.T.); 3Division of Epidemiology and Biostatistics, Department of Global Health, Stellenbosch University, Cape Town 7505, South Africa; 4Vaccine Preventable Diseases Programme, World Health Organization Regional Office for Africa, Brazzaville P.O. Box 06, Congo; 5Centre for Tropical Diseases and Global Health, Catholic University of Bukavu, Bukavu P.O. Box 285, Congo; 6Department of Community Health Studies, Faculty of Health Sciences, Durban University of Technology, Durban 4000, South Africa; 7Research, Development, Science and Innovation, Human Science Research Council, Durban 3209, South Africa; dsmith@hsrc.ac.za; 8Bill and Joyce Cummings Institute of Global Health, University of Global Health Equity (UGHE), Kigali P.O. Box 6955, Rwanda; potiku@ughe.org

**Keywords:** human papillomavirus, cervical cancer, vaccine confidence, vaccine safety, vaccine uptake, vaccination, Africa

## Abstract

**Background:** South Africa launched a school-based human papillomavirus (HPV) vaccination programme in 2014 and has achieved a national coverage of more than 80%. However, there is subnational variation in coverage, with eThekwini District in the province of KwaZulu-Natal having the lowest coverage at 40%. Knowledge of the factors associated with vaccine acceptance in this district would inform tailored strategies to improve coverage, which could be extrapolated to similar settings. We conducted this cross-sectional study to assess the factors associated with HPV vaccine acceptance in eThekwini District. **Methods:** We used stratified random sampling to select caregivers of children aged 9–14 years in the district. We interviewed participants in April–May 2023 and employed bivariate and multivariate logistic regression models to assess the factors associated with HPV vaccine acceptance. **Results:** Of 793 individuals contacted, 713 (89.9%) participated. Most were women (86.1%) and had a mean age of 42.6 ± 11.6 years and secondary or lower education (83.8%). Most participants knew about the HPV vaccination programme (86.0%) and accepted HPV vaccination (93.5%). The latter includes 42.9% who had already vaccinated their daughters and 50.6% who were willing to allow their daughters to be vaccinated. A negligible proportion was either undecided (2.1%) or unwilling (4.4%) to accept HPV vaccination. Awareness of the programme (adjusted odds ratio [aOR] 5.22; 95% confidence interval [95%CI] 2.01–13.56), confidence in vaccine safety (aOR 19.69; 95%CI 5.86–66.15), and endorsement by religious leaders (aOR 5.06; 95%CI 1.56–16.45) were independent predictors of vaccine acceptance. **Conclusions:** Our findings highlight the critical role of the provision of information and education about the benefits and safety of HPV vaccination.

## 1. Introduction

Human papillomavirus (HPV) encompasses a group of over 200 related viruses known to infect the genital areas, mouth, and throat of both men and women [1]. As a common sexually transmitted infection globally, HPV poses a significant health risk, being the primary cause of cervical cancer. Notably, HPV types 16 and 18 are implicated in around 70% of all cervical cancer cases worldwide [2,3]. The prevalence of HPV is so widespread that an estimated 80% of sexually active individuals will have been infected with at least one type of HPV by the age of 45 [3]. This high prevalence underscores the importance of early detection, regular screening, and preventive measures, notably vaccination, to mitigate the associated health risks [4,5].

The global fight against HPV has been bolstered by the development and dissemination of three primary HPV vaccines: the bivalent, quadrivalent, and nonavalent vaccines. These vaccines have demonstrated high safety, immunogenicity, and efficacy against the most oncogenic HPV types, especially types 16 and 18 [2]. The broad impact of these vaccines is seen in their ability to prevent specific HPV-related infections and conditions, as evidenced in the decreased rates of oral HPV infections among vaccinated young adults [6]. The administration of these vaccines typically occurs in a series of doses, with the recommended age for vaccination varying by country. The World Health Organization (WHO) advocates for achieving high vaccination coverage with at least one dose of an HPV vaccine among girls aged 9 to 14 years before they become sexually active, and up to three doses in people living with the human immunodeficiency virus [2].

In South Africa, cervical cancer ranks as a leading cause of cancer-related deaths among women, with the country experiencing a high number of new cases annually [7]. In response, South Africa launched a school-based HPV vaccination programme in 2014, targeting girls in grade four aged nine years or older, with the bivalent vaccine [8]. Since a substantial proportion of girls in grade four are below nine years of age, the target population for HPV vaccination in South Africa was later changed to grade five girls aged nine years or older [9]. While the programme has achieved significant vaccination coverage of more than 80% nationally, wide variations exist within and between subnational geographies, with some areas like the eThekwini District in the province of KwaZulu-Natal reporting vaccination coverage as low as 40% [8,9]. Research indicates that factors such as limited awareness, safety concerns, cultural beliefs, challenges with healthcare access, and social stigma around sexual health discussions contribute to these disparities in vaccine uptake [8,9,10]. Understanding these barriers and facilitators is crucial for designing targeted interventions to enhance vaccine uptake and, ultimately, reduce the burden of HPV-related diseases in South Africa. We therefore conducted this cross-sectional study to assess the behavioural and social factors that influence HPV vaccination decisions within the eThekwini District of KwaZulu-Natal Province in South Africa [10].

## 2. Materials and Methods

### 2.1. Study Design and Setting

This research, conducted between 15 April and 15 May 2023, took a close look at parents and caregivers in four communities within the eThekwini District: Chatsworth, Embo, Umlazi, and Wentworth. We chose a stratified random sampling method because it is proven to effectively ensure that we obtain a representative mix from different populations [11]. The district was divided into various layers based on sociodemographic factors like income, education, ethnicity, and whether people lived in urban or rural areas. This method aligns with the recommendations from Newman and Gough, who advocate for including a broad range of perspectives in educational and social research [12]. Each layer was represented in the sample in proportion to the district’s demographic makeup, allowing us to gain a well-rounded understanding of what influences the adoption of the HPV vaccine. We selected caregivers from various healthcare access points, including schools, public clinics, private practices, and community health centres. Additionally, we made sure to include caregivers with different job statuses and levels of community involvement to obtain a fuller picture of how community dynamics affect health behaviours. We worked closely with local healthcare providers, schools, and community organisations to recruit participants.

The decision to focus on caregivers was driven by several important factors, key among them being ethical and legal considerations. In South Africa, anyone under 18 is considered a minor, which means that studies involving them must follow strict ethical guidelines, like obtaining parental consent and the agreement of the adolescents themselves. These requirements can limit direct interaction with adolescents. Caregivers often play a crucial role in making health decisions for their children, including vaccinations. Their beliefs and attitudes significantly influence how the HPV vaccine is accepted. By concentrating on caregivers, we aimed to identify the barriers to important health interventions. Focusing on caregivers allowed us to identify challenges faced by key decision-makers, and including adolescents in the process would have required additional resources, like appropriate materials and measures to ensure confidentiality during data collection. Given the scope of the study, concentrating on caregivers was more practical. While this research emphasises caregivers, we also acknowledge the importance of hearing from adolescents. Future studies should include the perspectives of adolescent girls to gain a well-rounded understanding of HPV vaccine acceptance.

### 2.2. Sample Size

The survey was conducted on 713 caregivers of children in the age group of 9 to 14 years. This number was chosen to ensure sufficient statistical power to assess the determinants associated with HPV vaccination in the district [13]. This gives an 80% chance of detecting a small correlation effect (r = 0.2) at 5% significance level [13,14], meaning that even weak relationships in the data could be detected.

### 2.3. Sampling Procedure

We employed a stratified random sampling approach to ensure representation across the four distinct communities of Chatsworth, Embo, Umlazi, and Wentworth. Each of these locations was treated as a separate stratum, reflecting their unique geographic and sociodemographic characteristics. The total sample was proportionally allocated to each stratum based on available population estimates. Within each stratum, a sampling frame was developed using local administrative data provided by local authorities. From these lists, schools and health centres were randomly selected using a random number sequence. In each health centre and school, eligible respondents were randomly selected.

### 2.4. Data Management and Analyses

Trained research assistants played a crucial role in sharing the significance and procedures of the study with individuals who met the eligibility criteria. Those who agreed to participate signed an information leaflet and a consent form. After that, the trained research assistants used a structured questionnaire to gather the necessary data from the participants who consented. This survey was adapted from the WHO’s model for understanding the behavioural and social drivers of vaccination (BeSDs) [15]. We previously modified the BeSD questionnaire, which was originally created for childhood vaccinations, to explore COVID-19 vaccine acceptance in South Africa [16]. The BeSD framework is designed to investigate and measure various behavioural and social factors that affect vaccination decisions. It includes aspects like personal beliefs, trust, societal norms, motivation, and systemic obstacles to vaccine acceptance. By evaluating these factors, the BeSD model aids public health professionals and researchers in pinpointing the main motivators and barriers to vaccine uptake, ultimately helping to create more-effective and customised public health strategies to boost vaccine confidence and acceptance.

To ensure that our questionnaire was clear and relevant, we conducted pilot testing through cognitive interviews. This involved two rounds of interviews in the same location, with each round featuring four to eight participants. The goal of this pilot testing was to make sure that the participants could easily understand the questions and that their responses accurately reflected the concepts we intended to measure. This step was crucial to confirm that every survey item, including the questions and their answer choices, was properly translated and effectively conveyed the intended meanings about HPV vaccination. The revised questionnaire (see Appendix A) includes questions about sociodemographic characteristics, awareness of HPV, the link between HPV and cervical cancer, as well as information about the national HPV vaccination programme and its target audience. It also covers participants’ willingness to allow their daughters or close family members to receive the HPV vaccine, along with their perceptions of the vaccine’s safety. We summarised sociodemographic characteristics and knowledge and attitudes on HPV vaccination descriptively using frequencies for categorical variables and means and standard deviations for continuous variables. We applied logistic regression for both bivariate and multivariate analyses to examine the association between various factors and HPV vaccine acceptance, reporting odds ratios (ORs) with corresponding 95% confidence intervals (CIs) and p-values. Missing data were minimal across variables (<2%) and were handled using complete-case analysis. We conducted sensitivity checks to confirm that the exclusion of cases with missing values did not materially affect the distribution of the key characteristics or the model results. To explore potential effect modification, we tested interaction terms between key predictors, including awareness and education, and religious endorsement and information sources. None of the interaction terms reached statistical significance at the 0.05 level, and their inclusion did not improve the model fit as assessed by the Akaike Information Criterion (AIC); thus, they were excluded from the final model to maintain parsimony. Multicollinearity among the predictors was assessed using the Variance Inflation Factor (VIF), with all variables retained in the final model exhibiting VIF values below 2, indicating no significant collinearity. Given the high prevalence of vaccine acceptance in our sample, we acknowledge that the use of logistic regression may overestimate the strength of the associations. Nevertheless, we retained this approach for its interpretability and convergence stability. We used Microsoft Excel for our datasets and conducted analyses with the Statistical Package for Social Sciences (IBM SPSS Inc., Chicago, IL, USA) version 27.0 and R (The R Foundation for Statistical Computing, Vienna, Austria) version 4.0.5 software.

### 2.5. Ethics Approval

We obtained ethics approval from the University of Cape Town’s Human Research Ethics Committee (Reference: HREC 286/2021) and administrative clearance from the KwaZulu-Natal Provincial Department of Health.

## 3. Results

We invited 793 eligible participants, and 713 (89.9%) accepted the invitation, consented, and participated in the study. The characteristics of the study population are shown in Table 1. The mean age (±standard deviation) of the study population was 42.6 (±11.6) years, and there were 614 (86.1%) female participants. Most of the participants had secondary or lower education (83.7%) and household income less than ZAR 10,000 (South African Rand) (92.0%).

Among the participants, 70.9% (504/710) had heard of HPV, and 59.7% (425/712) knew that HPV causes cervical cancer. A higher proportion of 86.0% (602/700) were aware of the school-based HPV vaccination programme that targets girls in grade five in public schools. In addition, 93.7% (664/709) were confident that the HPV vaccine was safe for girls, and 77.0% (545/708) said the vaccine should also be offered to boys. Overall, 93.5% (667/713) of study participants accept HPV vaccination of their girls. This group includes 42.9% of study participants who had already vaccinated their daughters and 50.6% of participants who expressed willingness to allow their daughters or close relatives to receive the HPV vaccine. A negligible proportion was either undecided (2.1%) or unwilling to allow their daughters or next of kin to take the HPV vaccine (4.4%). The denominators differ slightly due to a small number of missing data.

Bivariate analyses revealed several factors to be significantly associated with the acceptance of HPV vaccination. These included good knowledge of HPV (crude OR 2.19; 95% CI 1.19 to 4.00; *p* = 0.011), recommendation from religious leaders to vaccinate (crude OR 8.99; 95% CI 3.39 to 23.81; *p* < 0.001), awareness of the school-based HPV vaccination programme (crude OR 6.96; 95% CI 3.67 to 13.17; *p* < 0.001), belief in the safety of vaccines (crude OR 26.89; 95% CI: 8.24 to 87.71; *p* < 0.001), and belief in the importance of vaccination (crude OR 3.37; 95% CI: 1.72 to 6.59; *p* < 0.001). Additionally, female caregivers had higher odds of HPV vaccine acceptance than that for male caregivers (crude OR 2.06; 95% CI 1.01 to 4.21; *p* = 0.047). As shown in Table 2, no significant statistical associations were found between HPV vaccine acceptance and education level, household income, age, or having received the COVID-19 vaccine.

In the multivariate analysis, which controlled for the sex of the caregiver, HPV knowledge, recommendation from religious leaders, awareness of the school-based HPV vaccination programme, confidence in vaccine safety, and belief in the importance of vaccination, three variables remained significantly associated with HPV vaccine acceptance. These were endorsement by religious leaders (adjusted OR 5.06; 95% CI 1.56 to 16.45; *p* = 0.007), awareness of the school-based HPV vaccination programme (adjusted OR 5.22; 95% CI 2.01–13.56; *p* = 0.001), and confidence in vaccine safety (adjusted OR 19.69; 95% CI 5.86 to 66.15; *p* < 0.001). These findings are displayed in Table 2 and Figure 1.

## 4. Discussion

This study examined caregiver knowledge, attitudes, and acceptance of the HPV vaccine in the eThekwini District of South Africa within the context of the national school-based vaccination programme. Despite high levels of acceptance (93.5%) and confidence in vaccine safety (93.7%), the actual uptake was significantly lower, with only 42.9% of caregivers reporting that their daughters had been vaccinated. Over 70% had heard of HPV, and nearly 60% knew of its link to cervical cancer. Awareness of the school-based vaccination programme was even more widespread (86%). Furthermore, confidence in the vaccine’s safety was strong, with 93.7% endorsing it as safe and 77% supporting its provision to boys. These figures suggest that national communication efforts have been partially successful in fostering positive attitudes toward vaccination. We found a disconnect between HPV vaccine acceptance and confidence in vaccine safety and the actual HPV vaccine uptake, which aligns with the well-documented “intention–behaviour gap” in the vaccination literature [17,18,19]. Applying the BeSD framework provides a more nuanced understanding of the interplay between cognitive, social, motivational, and structural factors influencing this gap. According to the BeSD framework, knowledge and positive beliefs alone are insufficient to ensure behaviour. Similar patterns have been observed in influenza and COVID-19 vaccination, where high intention did not translate into uptake due to psychological, contextual, or practical barriers [18,20]. These findings reinforce the need for strategies that convert awareness into action by targeting other behavioural determinants.

Social norms and trusted actors significantly shape vaccination decisions. Though not measured directly, the high levels of confidence reported by caregivers suggest that messages from schools, health authorities, and community health workers are largely trusted. This mirrors findings from other LMICs where trusted voices such as teachers, nurses, and community leaders are critical in reinforcing vaccine acceptance [21,22]. A particularly salient insight from this study is the influence of religious endorsement on vaccination decisions. Caregivers who perceived support from religious leaders were more likely to accept HPV vaccination for their children. This aligns with patterns observed in sub-Saharan Africa where religious authority figures hold considerable sway in shaping health behaviours [23]. Religious institutions can serve as trusted intermediaries, particularly where formal health systems face credibility gaps. Structuring partnerships with faith leaders—through endorsements, sermon messaging, or faith-based outreach—offers a viable and culturally sensitive strategy for reinforcing vaccine confidence and bridging the intention–behaviour gap.

Motivation to vaccinate appeared high in this study, reflected in the large proportion of caregivers expressing willingness to vaccinate their daughters. However, actual vaccination was significantly lower, indicating that practical constraints may have interfered. The BeSD framework highlights that logistical issues such as lack of reminders, difficulty returning consent forms, or unclear scheduling can inhibit motivated individuals from acting [24]. This was evident in eThekwini, where even supportive caregivers may have encountered barriers related to timing, communication breakdowns, or lack of follow-up by school health teams. Prior works also supports this, showing that convenience, cost, and uncertainty are persistent obstacles to vaccine completion [19,20].

The findings from eThekwini indicate critical leverage points for reinforcing the HPV vaccination programme. Vaccine literacy and trust were strong, but behavioural outcomes lagged. Bridging this gap requires system-level changes. Enhancing school communication systems through reminders, SMS alerts, or digital tracking of consent forms could streamline the pathway from intention to uptake. Moreover, engaging fathers, religious leaders, and other community influencers could enhance collective decision-making and strengthen vaccine uptake. Social media also presents a promising avenue for expanding knowledge and shaping norms. Evidence suggests that culturally tailored digital campaigns can amplify pro-vaccine messaging and counter misinformation [25]. The integration of digital and face-to-face strategies could form a comprehensive approach to community engagement.

### 4.1. Implications and Recommendations

The findings of this study highlight several critical areas for future research to strengthen HPV vaccination strategies in eThekwini. While caregiver awareness and vaccine confidence were high, actual vaccine uptake lagged significantly behind, reinforcing the importance of investigating the intention–behaviour gap more systematically [17,19]. Future studies should use longitudinal designs to track how behavioural intentions translate into actual vaccination behaviour over time and in response to health system interventions.

Further, although this study applied the BeSD framework to interpret findings, additional research is needed to directly measure the psychological and structural determinants of HPV vaccination. Mixed-methods research that includes the qualitative exploration of motivational drivers, perceived barriers, and community dynamics would enrich current understanding. This is particularly important in light of the high caregiver endorsement of vaccine safety and acceptance yet limited follow-through. Investigating the role of healthcare providers, school staff, and religious or community leaders in shaping social norms and decision-making could yield actionable insights [21].

From a policy perspective, the school-based HPV vaccination programme in eThekwini is grounded in a sound structure but requires strategic enhancements to maximise its potential impact. Training and empowering educators and school nurses to communicate confidently about HPV vaccination should be prioritised. This addresses Xu et al.’s (2024) observation that low vaccine literacy and awareness, particularly about cancer prevention, remain key barriers in low- and middle-income countries (LMICs) [22]. Strengthening this frontline capacity can improve informed decision-making at the community level [22].

Kutz et al. (2023) emphasised the importance of stakeholder engagement and tailored health education in sub-Saharan Africa [23]. In eThekwini, this supports the recommendation to invest in culturally relevant, community-driven mobilisation strategies, including partnerships with religious leaders, who can act as trusted influencers. Such endorsements could be formally incorporated into outreach campaigns to increase credibility and local resonance.

The strong caregiver support for gender-neutral vaccination suggests both social readiness and policy alignment with WHO recommendations [22]. Expanding the programme to include boys could reduce stigma and normalise vaccination among all adolescents. Research should assess whether this expansion influences parental attitudes and improves overall coverage.

Popa et al. (2022) highlight that logistical barriers and geographic access constraints disproportionately affect vaccine uptake in underserved areas [26]. Accordingly, deploying mobile vaccination units and decentralised service points in eThekwini could address access-related inequities. Evaluating these outreach models would inform their scalability and effectiveness.

Ortiz et al. (2019) demonstrated the potential of digital platforms to increase HPV awareness and correct misinformation [25]. In this context, integrating social media campaigns with school- and community-based efforts could enhance vaccine confidence and visibility. Future research should assess which digital strategies most effectively engage caregivers and adolescents in South Africa’s diverse sociocultural landscape.

Lorini et al. (2022) found that reliance on trusted health sources such as general practitioners was associated with higher vaccine literacy [27]. This insight supports the inclusion of local healthcare workers in mobilisation campaigns, equipping them with materials and messaging that reinforce school-based efforts. Simultaneously, vaccine literacy interventions should be developed and tested within South African settings to promote critical engagement with vaccination information.

Finally, the heterogeneity of HPV vaccine uptake across South Africa necessitates context-sensitive planning [8]. Future studies should map regional disparities and trial locally adapted interventions to identify what works, for whom, and under what conditions. Mixed-methods approaches will be especially valuable in unpacking the interplay between cultural, structural, and psychological drivers of vaccine behaviour.

### 4.2. Strengths and Limitations

This study provides valuable insights into caregiver awareness, attitudes, and acceptance of HPV vaccination within the school-based programme in eThekwini District. A key strength lies in the application of the WHO BeSD framework, which enabled a multidimensional analysis of behavioural and social drivers. The large sample size (n = 713) and high response rate (90%) enhance the internal validity of our findings, while the focus on both vaccine literacy and confidence offers a nuanced understanding of caregiver decision-making.

However, several limitations must be noted. First, reliance on self-reported data may introduce recall or social desirability bias, potentially inflating acceptance estimates. Triangulation with administrative data would help quantify the intention–behaviour gap more precisely. Second, missing data were minimal (<2%) and handled via complete-case analysis, which may slightly affect representativeness.

Third, the urban setting and the predominance of female respondents (86.1%) may limit generalisability to rural or underserved populations. Fourth, although logistic regression was appropriate, it may overestimate associations due to the high prevalence of acceptance; alternative models could be explored in future analyses. Finally, while the BeSD framework guided the interpretation, we did not directly measure some structural and psychological barriers. Future studies should incorporate mixed-methods or longitudinal designs to capture these dimensions more comprehensively.

In summary, combining behavioural theory with robust methodology and community engagement is essential for designing effective and equitable HPV vaccination strategies in South Africa and similar settings.

## 5. Conclusions

Despite the high awareness and acceptance of the HPV vaccine among caregivers in eThekwini District, the actual uptake remains low. Using the BeSD framework, we identified gaps driven by social, motivational, and logistical barriers. Knowledge and confidence alone are not sufficient to ensure vaccine coverage.

Bridging this gap requires stronger school-based delivery, improved follow-through, and the engagement of trusted community voices. To reduce cervical cancer burden and achieve the WHO 90–70–90 targets, HPV vaccination strategies must integrate vaccine literacy, system-level support, and community engagement.

## Figures and Tables

**Figure 1 vaccines-13-00732-f001:**
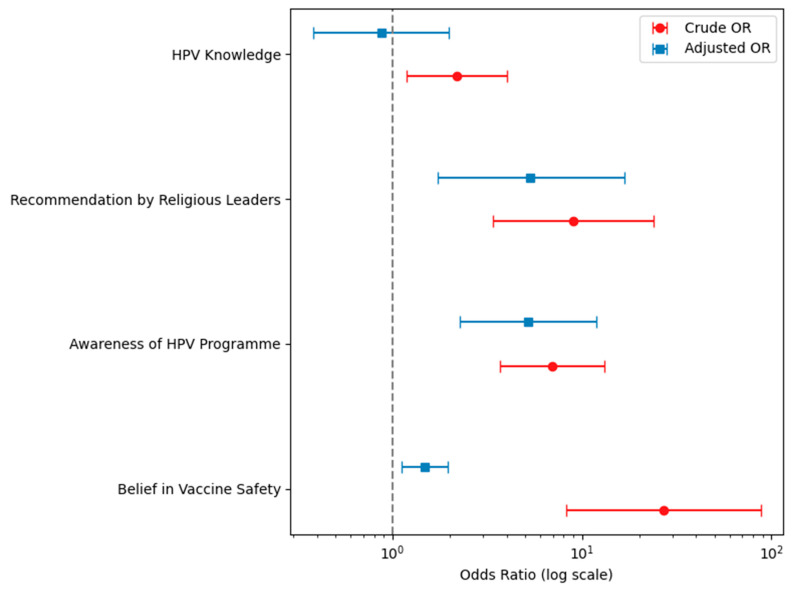
Crude and adjusted odds ratios of willingness to take up HPV vaccination by HPV knowledge and attitudes. The figure displays both crude and adjusted odds ratios (ORs) with 95% confidence intervals (CIs) for key factors associated with HPV vaccine uptake. These factors include knowledge of HPV, awareness of the school-based HPV vaccination programme, perceived importance and safety of the vaccine, and recommendations from religious leaders. Blue squares represent crude ORs, and red circles represent adjusted ORs from multivariable models. The vertical dashed grey line at OR = 1.0 denotes the reference category for each variable. Adjusted models accounted for potential confounders. ORs greater than 1 indicate a higher odds of acceptance of vaccine uptake compared with that in the reference group.

**Table 1 vaccines-13-00732-t001:** Characteristics of the study population.

Exploratory Variables	Summary Statistics		
	All	Accept HPV Vaccine	Do Not Accept HPV Vaccine
Sample size	N = 713 (100%)	N = 667 (93.5%)	N = 46 (6.5%)
Mean age (standard deviation)	42.6 (11.6) years	42.5 (11.4) years	44.5 (12.9) years
Sex: Male Female	99 (13.9%) 614 (86.1%)	88 (13.2%) 579 (86.8%)	11 (23.9) 35 (76.1)
Education: Primary and below Secondary Tertiary and above Missing	45 (6.3%) 552 (77.4%) 113 (15.8%) 5 (0.4%)	44 (6.6%) 513 (76.9%) 105 (15.7%) 5 (0.7%)	1 (2.2%) 38 (82.6%) 7 (15.2%) 0 (0.0%)
Household income: Less than ZAR 10,000 * ZAR 10,000 to 20,000 More than ZAR 20,000	656 (92.0%) 41 (5.8%) 16 (2.2%)	613 (91.9%) 38 (5.7%) 14 (2.1%)	41 (89.1%) 3 (6.52) 2 (4.3%)
Neighbourhood: Chatsworth Embo Wentworth Umlazi Missing	182 (25.5%) 144 (20.2%) 190 (26.6%) 197 (27.6%) 2 (0.3%)	169 (25.3%) 134 (20.1%) 166 (24.9%) 196 (29.4%) 2 (0.3%)	13 (28.3%) 10 (21.7%) 23 (50.0%) 0 (0.0%) 0 (0.0%)

* ZAR = South African Rand (currently, USD 1 (United States Dollar) = ZAR 18).

**Table 2 vaccines-13-00732-t002:** Predictors of acceptance of HPV in eThekwini District of KwaZulu-Natal Province, South Africa.

Variable	N	Crude OR	95% CI	*p*-Value	Adjusted OR	95% CI	*p*-Value
Sex of caregiver	713						
Male	99	Baseline					
Female	614	2.06	1.01–4.21	0.047	0.78	0.32–1.89	0.588
Education	710				–	–	–
Primary	45						
Secondary	552	0.31	0.04–2.29	0.249			
Tertiary	113	0.34	0.04–2.85	0.321			
Household income	713				–	–	–
<ZAR 3000	541	Baseline					
ZAR 3000–10,000	115	1.26	0.52–3.07	0.609			
ZAR 100,001–20,000	41	0.88	0.26–2.99	0.837			
>ZAR 20,000	16	0.49	0.11–2.22	0.353			
Age	711	0.99	0.96–1.01	0.254	–	–	–
Received COVID-19 vaccine	712						
No	260	Baseline					
Yes	252	1.42	0.77–2.60	0.262	–	–	–
HPV knowledge	710						
Poor	206	Baseline					
Good	504	2.19	1.19–4.00	0.011	0.84	0.33–2.11	0.710
Recommendation of religious leaders on vaccination	711						
Discourage	20	Baseline					
Encourage	691	8.99	3.39–23.81	<0.001	5.06	1.56–16.45	0.007
Aware of school-based HPV vaccination programme	700						
No	98	Baseline					
Yes	602	6.96	3.67–13.17	<0.001	5.22	2.01–13.56	0.001
View on safety of vaccines	708						
Sceptical	90	Baseline					
Safe	618	26.89	8.24–87.71	<0.001	19.69	5.86–66.15	<0.001
View on importance of vaccines	708						
Not important	90	Baseline					
Important	618	3.37	1.72–6.59	<0.001	1.20	0.32–1.89	0.710

aOR: adjusted odds ratio; cOR: crude odds ratio; CI: confidence interval; ZAR: South African Rand (currently, USD 1 (United States Dollar) is equivalent to about ZAR 18).

## Data Availability

The data generated in this study is available from the corresponding author upon reasonable request.

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
