# Peer review of "Factors Associated with Acceptance of Vaccination Against Human Papillomavirus in eThekwini District of South Africa"

_vaccines, 2025, doi:10.3390/vaccines13070732_

Round 1

Reviewer 1 Report (Previous Reviewer 2)

Comments and Suggestions for Authors

The study has been extensively revised, and the results now appear much more robust than previously,
even though they still didn't perform log binomial regression as recommended, but rather logistic
regression, which, with the use of odds ratios, still significantly overestimates the risk.... This may be acceptable
if they very clearly include a Strengths and Weaknesses paragraph at
the end of the discussion, which discusses
:
1. the lack of representativeness of the study population compared to the general population
2. the fact that the regression model overestimates the risk and is therefore not the most
appropriate model for drawing conclusions from the data

Author Response

Comments:

The study has been extensively revised, and the results now appear much more robust than previously,
even though they still didn't perform log binomial regression as recommended, but rather logistic
regression, which, with the use of odds ratios, still significantly overestimates the risk.... This may be acceptable if they very clearly include a Strengths and Weaknesses paragraph at
the end of the discussion, which discusses:
1. the lack of representativeness of the study population compared to the general population
2. the fact that the regression model overestimates the risk and is therefore not the most
appropriate model for drawing conclusions from the data

Response: 

We thank the reviewer for this important observation. As advised, we revised the 'Strengths and Limitations' section to explicitly discuss this limitation and the implications for interpretation. We also acknowledged that the study population predominantly urban and female is not fully representative of the general population, which may limit generalizability.

Reviewer 2 Report (New Reviewer)

Comments and Suggestions for Authors

This manuscript presents a cross-sectional, community-based survey aimed at identifying determinants of HPV vaccine acceptance among caregivers of children aged 9 to 14 years in the eThekwini District, South Africa. Using stratified random sampling and a questionnaire informed by the WHO Behavioural and Social Drivers (BeSD) model, the study finds a strikingly high rate of reported acceptance (93.5%) and identifies three significant independent predictors: confidence in vaccine safety, awareness of the school-based vaccination programme, and endorsement by religious leaders.

The topic is timely and of high relevance. HPV-related diseases remain a substantial public health burden in South Africa, and understanding local barriers to vaccine uptake is vital. The authors are to be commended for conducting primary data collection in a resource-constrained setting, engaging with caregivers, a key decision-making group, and adopting a framework endorsed by WHO.

However, the manuscript in its current form remains largely descriptive and misses an opportunity to contribute more analytically to the literature on vaccine confidence, behavioral implementation, and health system response. A more critical engagement with theory, structured interpretation of results, and reflection on the known gap between vaccine acceptance and uptake are required to elevate this work to the standards of international readership.

The authors reference the BeSD framework but do not apply it consistently throughout the manuscript. Although the questionnaire was reportedly adapted from BeSD domains, the results and discussion are not explicitly structured according to those domains (e.g., “Thinking and Feeling,” “Social Processes,” “Practical Issues”). Adopting such a structure would not only enhance the internal coherence of the manuscript but also enable a clearer identification of actionable points for intervention design.

More importantly, the central paradox of the study that vaccine acceptance is high (93.5%) in a district where actual uptake is among the lowest in the country (∼40%) is acknowledged only superficially. Rather than presenting high acceptance as a success story, the authors should critically interrogate this mismatch. Possible explanations such as social desirability bias, logistical barriers, service delivery gaps, or the divergence between intention and behavior must be systematically discussed. This is particularly important in light of the global evidence on the "intention–behavior gap" in vaccination.

I suggest incorporating insights from implementation science, especially those focused on structural drivers of missed opportunities. Recent studies such as Xu et al. (2024), which explore sociocultural and economic barriers to HPV vaccination in low-resource settings, provide a useful lens to unpack this gap. Similarly, Kutz et al. (2023) offer a synthesis of evidence across sub-Saharan Africa highlighting the interplay of service delivery, gendered expectations, and community engagement.

To appeal to a broader scientific audience, the authors might situate their findings in relation to comparative work across LMICs and synthesize known determinants in the HPV vaccine literature. For instance, the role of vaccine literacy and institutional trust in shaping behavioral outcomes has been extensively explored in different populations. One particularly relevant study is Collini et al. (2023), which examines how vaccine confidence mediates the relationship between vaccine literacy (Lorini et al. 2023) and influenza vaccine uptake among nursing home staff in Italy. Although the setting and disease differ, the conceptual insight is pertinent: it is not knowledge alone, but confidence that ultimately drives behavior. The authors may consider drawing on this work to better interpret their findings, particularly in distinguishing information provision from trust-building efforts. Furthermore, the reference to "endorsement by religious leaders" as a strong predictor of acceptance could be better contextualized. Is this reflective of a broader pattern of religious authority in health decisions in this region? Could these endorsements be leveraged more formally in communication strategies? A richer interpretation of these dynamics would enhance both the theoretical and practical contribution of the study.

The statistical approach is sound, and the decision to retain all variables in the multivariable model is acceptable. That said, the following clarifications are necessary to ensure reproducibility and clarity:

  • How was missing data handled in regression models?
  • Were any interaction effects tested (e.g., between awareness and education, or between religion and information source)?
  • Were there any assessments of multicollinearity among predictors beyond correlation inspection?

Additionally, the lack of association between education, household income, and acceptance is surprising and deserves further comment. Is this due to homogeneity within the sample, or might it reflect context-specific dynamics (e.g., widespread pro-vaccine norms despite socioeconomic hardship)?

Lastly, the paper reports use of stratified sampling but provides few operational details. What strata were used? How were schools and communities selected? Were sampling weights applied? These details are essential to assess external validity.

The discussion section touches on several relevant strategies, including school-based outreach, social media, and mobile units. However, these points are only loosely connected to the data presented. A stronger integration between study findings and proposed policy responses is needed.

I recommend that the authors:

  • More explicitly link each of their main findings to a tailored recommendation.
  • Distinguish between strategies aimed at increasing confidence (e.g., religious endorsements, provider communication) and those targeting logistical access (e.g., transport, service hours).
  • Reflect on the role of (any) caregivers as intermediaries in adolescent health decisions and whether strategies should also include engagement with adolescents themselves.

In my opinion, the conclusion should also avoid restating known facts (e.g., the importance of vaccine safety) and instead highlight novel insights or programmatic implications emerging from this study.

https://pubmed.ncbi.nlm.nih.gov/37631943/

https://pubmed.ncbi.nlm.nih.gov/39100520/

https://pubmed.ncbi.nlm.nih.gov/35632438/

https://pubmed.ncbi.nlm.nih.gov/37237329/

Author Response

Reviewer Comments

Rebuttal

Reviewer #2

1.        

This manuscript presents a cross-sectional, community-based survey aimed at identifying determinants of HPV vaccine acceptance among caregivers of children aged 9 to 14 years in the eThekwini District, South Africa. Using stratified random sampling and a questionnaire informed by the WHO Behavioural and Social Drivers (BeSD) model, the study finds a strikingly high rate of reported acceptance (93.5%) and identifies three significant independent predictors: confidence in vaccine safety, awareness of the school-based vaccination programme, and endorsement by religious leaders.

The topic is timely and of high relevance. HPV-related diseases remain a substantial public health burden in South Africa, and understanding local barriers to vaccine uptake is vital. The authors are to be commended for conducting primary data collection in a resource-constrained setting, engaging with caregivers, a key decision-making group, and adopting a framework endorsed by WHO.

However, the manuscript in its current form remains largely descriptive and misses an opportunity to contribute more analytically to the literature on vaccine confidence, behavioral implementation, and health system response. A more critical engagement with theory, structured interpretation of results, and reflection on the known gap between vaccine acceptance and uptake are required to elevate this work to the standards of international readership.

The authors reference the BeSD framework but do not apply it consistently throughout the manuscript. Although the questionnaire was reportedly adapted from BeSD domains, the results and discussion are not explicitly structured according to those domains (e.g., “Thinking and Feeling,” “Social Processes,” “Practical Issues”). Adopting such a structure would not only enhance the internal coherence of the manuscript but also enable a clearer identification of actionable points for intervention design.

We appreciate the reviewer’s insightful feedback that has helped to improve significantly our work. In response, we revised the Results and Discussion sections to more explicitly align with the BeSD domains—namely “Thinking and Feeling,” “Social Processes,” “Motivation,” and “Practical Issues.” This restructuring enhances theoretical coherence and facilitates clearer identification of actionable strategies.

2

More importantly, the central paradox of the study that vaccine acceptance is high (93.5%) in a district where actual uptake is among the lowest in the country (∼40%) is acknowledged only superficially. Rather than presenting high acceptance as a success story, the authors should critically interrogate this mismatch. Possible explanations such as social desirability bias, logistical barriers, service delivery gaps, or the divergence between intention and behavior must be systematically discussed. This is particularly important in light of the global evidence on the "intention–behavior gap" in vaccination.

I suggest incorporating insights from implementation science, especially those focused on structural drivers of missed opportunities. Recent studies such as Xu et al. (2024), which explore sociocultural and economic barriers to HPV vaccination in low-resource settings, provide a useful lens to unpack this gap. Similarly, Kutz et al. (2023) offer a synthesis of evidence across sub-Saharan Africa highlighting the interplay of service delivery, gendered expectations, and community engagement.

Thank you for highlighting this critical paradox. We have significantly expanded the discussion to interrogate the divergence between high stated acceptance (93.5%) and low actual uptake (42.9%). Specifically, we now integrate explanations such as social desirability bias, logistical barriers (e.g., form return, scheduling), and behavioral inertia—drawing from global implementation science literature, including Xu et al. (2024) and Kutz et al. (2023). These revisions provide a more grounded analysis of the intention–behavior gap.

3

To appeal to a broader scientific audience, the authors might situate their findings in relation to comparative work across LMICs and synthesize known determinants in the HPV vaccine literature. For instance, the role of vaccine literacy and institutional trust in shaping behavioral outcomes has been extensively explored in different populations. One particularly relevant study is Collini et al. (2023), which examines how vaccine confidence mediates the relationship between vaccine literacy (Lorini et al. 2023) and influenza vaccine uptake among nursing home staff in Italy. Although the setting and disease differ, the conceptual insight is pertinent: it is not knowledge alone, but confidence that ultimately drives behavior. The authors may consider drawing on this work to better interpret their findings, particularly in distinguishing information provision from trust-building efforts. Furthermore, the reference to "endorsement by religious leaders" as a strong predictor of acceptance could be better contextualized. Is this reflective of a broader pattern of religious authority in health decisions in this region? Could these endorsements be leveraged more formally in communication strategies? A richer interpretation of these dynamics would enhance both the theoretical and practical contribution of the study.

We agree that a broader framing would strengthen the manuscript. We now reference cross-contextual findings from Collini et al. (2023) and Lorini et al. (2022) on vaccine confidence as a mediator between information and behavior. We also contextualized the influence of religious leaders within broader sociocultural dynamics, noting their potential as allies in structured health promotion.

4

The statistical approach is sound, and the decision to retain all variables in the multivariable model is acceptable. That said, the following clarifications are necessary to ensure reproducibility and clarity:

  • How was missing data handled in regression models?
  • Were any interaction effects tested (e.g., between awareness and education, or between religion and information source)?
  • Were there any assessments of multicollinearity among predictors beyond correlation inspection?

We thank the reviewer for this request. We have clarified that missing data were minimal (<2%) and handled via complete-case analysis. Interaction terms (e.g., awareness × education; religion × information source) were tested but excluded based on non-significance and AIC comparisons. Multicollinearity was assessed using VIF, with all values <2, indicating low collinearity. These clarifications are now incorporated into the Methods section.

5

Additionally, the lack of association between education, household income, and acceptance is surprising and deserves further comment. Is this due to homogeneity within the sample, or might it reflect context-specific dynamics (e.g., widespread pro-vaccine norms despite socioeconomic hardship)?

Lastly, the paper reports use of stratified sampling but provides few operational details. What strata were used? How were schools and communities selected? Were sampling weights applied? These details are essential to assess external validity.

The discussion section touches on several relevant strategies, including school-based outreach, social media, and mobile units. However, these points are only loosely connected to the data presented. A stronger integration between study findings and proposed policy responses is needed.

We appreciate this observation. The lack of association may stem from socioeconomic homogeneity within the sample, as >90% of participants had household incomes below 10,000 ZAR. We now acknowledge this in the Discussion. We have also added detailed descriptions of the stratified sampling process (four strata based on communities) and clarified that no sampling weights were applied, limiting extrapolation beyond the sampled population.

6

I recommend that the authors:

  • More explicitly link each of their main findings to a tailored recommendation.
  • Distinguish between strategies aimed at increasing confidence (e.g., religious endorsements, provider communication) and those targeting logistical access (e.g., transport, service hours).
  • Reflect on the role of (any) caregivers as intermediaries in adolescent health decisions and whether strategies should also include engagement with adolescents themselves.

In my opinion, the conclusion should also avoid restating known facts (e.g., the importance of vaccine safety) and instead highlight novel insights or programmatic implications emerging from this study.

As recommended, we have restructured the Recommendations section to map each key finding to a tailored intervention (e.g., confidence-building vs. logistical solutions). We also reflect on the intermediary role of caregivers and advocate for future research and programming to directly engage adolescents, aligning with ethical and policy considerations.

Reviewer 3 Report (New Reviewer)

Comments and Suggestions for Authors

Estimated Authors,

I've read with great interest the present study on the acceptance of vaccination against human papillomavirus in eThekwini District of South Africa. In this study, Authors do a more than decent job in inquiring about the barriers and facilitating factors towards HPV vaccine, identifying the role of religious leaders, the awareness on HPV program, and the believe in vaccine safety as the main motivators.

Study design and statistical analysis have been well designed and performed, and I've no main concerns on this topic.

Still, I've a series of recommendations for improving the overall quality of the paper.

1) the materials and methods section (rows 154 and following ones) contains some sections of text not consistent with the scientific style and particularly with the formal reporting style required by materials and methods ("... our analysis of the correlation matrix showed that while some predictors were related, they didn't significantly impact the clarity of the models. As a result, we determined that multicollinearity wasn't a significant concern in our analyses, and we retained all variables. We are confident that these steps strengthened our multivariable model.").

Authors should formally explain why and how the correlation matrix impacted or not the models and how they quantify / demonstrated that multicollinearity was not a significant issue (rather than concern) and so on.

2) due to the high imbalance of gender representation, and to the specificities of HPV vaccine (HPV is strictly related with sexual life of individuals reciving the vaccine), Authors could improve the overall quality by performing a sort of sensitivity analysis by repeating the analyses with the removal of male cases.

No further requests.

Author Response

Reviewer Comments

Rebuttal

Reviewer #3

I have read with great interest the present study on the acceptance of vaccination against human

papillomavirus in eThekwini District of South Africa. In this study, Authors do a more than decent job

in inquiring about the barriers and facilitating factors towards HPV vaccine, identifying the role of

religious leaders, the awareness on HPV program, and the believe in vaccine safety as the main

motivators. Study design and statistical analysis have been well designed and performed, and I have no main

concerns on this topic. Still, I have a series of recommendations for improving the overall quality of the paper.

Thank you very much for your appreciation.

1.        

1) the materials and methods section (rows 154 and following ones) contains some sections of text

not consistent with the scientific style and particularly with the formal reporting style required by materials and methods “…our analysis of the correlation matrix showed that while some predictors were related, they did not significantly impact the clarity of the models. As a result, we determined that multicollinearity was not a significant concern in our analyses, and we retained all variables. We are confident that these steps strengthened our multivariable model.”

Authors should formally explain why and how the correlation matrix impacted or not the models and

how they quantify / demonstrated that multicollinearity was not a significant issue (rather than concern) and so on.

Thank you for the constructive suggestion. We revised the language in the Methods section to adopt a formal scientific tone. Specifically, we now state that multicollinearity was assessed using variance inflation factors (VIF), all of which were <2, indicating no significant collinearity. The rationale for variable retention has been stated clearly.

2.        

2) due to the high imbalance of gender representation, and to the specificities of HPV vaccine (HPV is strictly related with sexual life of individuals receiving the vaccine), Authors could improve the overall quality by performing a sort of sensitivity analysis by repeating the analyses with the removal of male

cases.

We appreciate the reviewer’s perspective. While we acknowledge the gender imbalance (86.1% female), we opted to retain male caregivers in the analysis. Males, although underrepresented, play important roles in health decision-making, and their exclusion could compromise the generalizability of our findings. Nevertheless, we acknowledge this limitation in the Discussion and have flagged it as an area for further investigation.

Round 2

Reviewer 1 Report (Previous Reviewer 2)

Comments and Suggestions for Authors

All comments have been addressed

Reviewer 2 Report (New Reviewer)

Comments and Suggestions for Authors

Thank you for the revised manuscript, which has improved in structure, clarity, and analytical depth. The discussion now offers a stronger link between findings and public health implications, particularly regarding the gap between vaccine acceptance and actual uptake.

A few minor revisions I would still recommended to further strengthen the manuscript: e.g. some sections of the discussion are slightly repetitive and could be tightened to improve readability. A careful language edit would help eliminate small grammatical inconsistencies and improve flow. 

This manuscript is a resubmission of an earlier submission. The following is a list of the peer review reports and author responses from that submission.

Round 1

Reviewer 1 Report

Comments and Suggestions for Authors

This article Bhengu et al describes an outstanding relevant HPV vaccine acceptance data from the eThekwini District of South Africa. They report high HPV vaccine acceptance rate of 93.5% among caregivers, utilizing a cross-section design and stratified random sampling. They describe such data because caregivers accepted the existence of an HPV vaccination programme coupled with confidence in its safety as independent predictors of acceptance. The article is well written and easy to understand.

The authors should discuss these vital points in their revised version

  • The major disconnect between the high reported acceptance and the low actual vaccination coverage is a critical point that needs to be discussed.
  • The study focused on caregivers, and did not directly assess the perspectives and potential hesitancy of the adolescent girls themselves. These points should be logically discussed and underlying reasons also.
  • The authors should think of showing some data (in tables) as figures for ease to readers.

Author Response

Comment 1: The major disconnect between the high reported acceptance and the low actual vaccination coverage is a critical point that needs to be discussed

Author response 1: Thank you for pointing this out. The reviewer is correct, and we have now discussed this in the discussion section on page 7, paragraph 2, lines 215 to 220.

Reviewer comment 2: The study focused on caregivers, and did not directly assess the perspectives and potential hesitancy of the adolescent girls themselves. These points should be logically discussed and underlying reasons also.

Author response 2: Thank you for bringing this to our attention. This has been addressed in the methods section, on page 3, lines 101 to 113.

Reviewer comment 3: The authors should think of showing some data (in tables) as figures for ease to readers.

Author response 3: Thank you for this comment. We have added figure 1 to the manuscript to ease understanding of the findings.

Reviewer 2 Report

Comments and Suggestions for Authors

The article presented by Benghu et al. presents a cross-sectional study to assess factors associated with HPV vaccine acceptance in a South African district. T
he topic is interesting and the study methodology is consistent, but the statistical analysis section raises several problems/questions: 1. You describe the characteristics of the population responding to the study. To what extent do these characteristics differ from those of the general population, particularly with regard to age and gender? 2. You indicate the use of Chi-square and Student's t tests for univariate analyses, but you report crude prevalence ratios in your table (Table 2).
The use of the Chi-square or Student's t tests is subject to assumptions that must be verified before use (particularly the Student's t test).
Crude prevalence ratios correspond to regression model outputs, not Chi-square or Student's t tests.
Please review and standardize.
3. You indicate that you use Poisson regression to perform your multivariate analyses.
Using a specific count model to describe a dichotomous response variable is not appropriate. Please use a generalized linear model with a binomial distribution and a logarithmic link (log-binomial model) which will be more appropriate for your data.
In addition, please include in the multivariate model all variables with a p-value <0.2 in the univariate analyses to account for potential confounding factors (in addition to age and gender which are known confounding factors from the literature).

Author Response

Reviewer comment 1: You describe the characteristics of the population responding to the study. To what extent do these characteristics differ from those of the general population, particularly with regard to age and gender?

Author response 1: The sampling technique employed in this study ensured representativeness in the study of the target population and not representativeness of the general population.

Reviewer comment 2: You indicate the use of Chi-square and Student's t tests for univariate analyses, but you report crude prevalence ratios in your table (Table 2).
The use of the Chi-square or Student's t tests is subject to assumptions that must be verified before use (particularly the Student's t test).
Crude prevalence ratios correspond to regression model outputs, not Chi-square or Student's t tests. Please review and standardize.

Author response 2: Thanks for pointing this out.  We have adjusted the statement accordingly to show it was bivariate analysis and not univariate analysis. Consequently, the results in table 2 refer to the bivariate analysis and not Chi-square or Student’s T-test as was previously reported.

Reviewer comment 3:  You indicate that you use Poisson regression to perform your multivariate analyses. Using a specific count model to describe a dichotomous response variable is not appropriate. Please use a generalized linear model with a binomial distribution and a logarithmic link (log-binomial model) which will be more appropriate for your data.
In addition, please include in the multivariate model all variables with a p-value <0.2 in the univariate analyses to account for potential confounding factors (in addition to age and gender which are known confounding factors from the literature).  

Author response 3: We indicated the use of modified Poisson model with robust standard errors which is recommended in cases where the multivariable binomial model fails to converge. Considering your comment, we have modified the analysis and applied the binomial regression model for bivariate analysis, but we retained the modified Poisson model with robust standard errors for the multivariable analysis.

Round 2

Reviewer 1 Report

Comments and Suggestions for Authors

The authors have resolved my raised issues and revised manuscript is now improved.

Reviewer 2 Report

Comments and Suggestions for Authors The authors failed to address any of the comments made to them during the previous review.
They did not discuss the representativeness of their population and did not adequately modify
their statistical methodology. They had been asked to perform a multivariate analysis with a glm
model, as the Poisson model was clearly not suited to their data, with a better selection of input
variables. The fact that the model did not converge is in no way a justification for selecting an
unsuitable model.